# A Circum-Arctic Monitoring Framework for Quantifying Annual Erosion Rates of Permafrost Coasts

**Marius Philipp** [1,2,*], **Andreas Dietz** [2], **Tobias Ullmann** [1] and **Claudia Kuenzer** [1,2]

1    Department of Remote Sensing, Institute of Geography and Geology, University of Würzburg, Am Hubland, 97074 Würzburg, Germany
2    German Remote Sensing Data Center (DFD), German Aerospace Center (DLR), Muenchener Strasse 20, 82234 Wessling, Germany
*    Correspondence: marius.philipp@uni-wuerzburg.de

**Abstract:** This study demonstrates a circum-Arctic monitoring framework for quantifying annual change of permafrost-affected coasts at a spatial resolution of 10 m. Frequent cloud coverage and challenging lighting conditions, including polar night, limit the usability of optical data in Arctic regions. For this reason, Synthetic Aperture RADAR (SAR) data in the form of annual median and standard deviation (sd) Sentinel-1 (S1) backscatter images covering the months June–September for the years 2017–2021 were computed. Annual composites for the year 2020 were hereby utilized as input for the generation of a high-quality coastline product via a Deep Learning (DL) workflow, covering 161,600 km of the Arctic coastline. The previously computed annual S1 composites for the years 2017 and 2021 were employed as input data for the Change Vector Analysis (CVA)-based coastal change investigation. The generated DL coastline product served hereby as a reference. Maximum erosion rates of up to 67 m per year could be observed based on 400 m coastline segments. Overall highest average annual erosion can be reported for the United States (Alaska) with 0.75 m per year, followed by Russia with 0.62 m per year. Out of all seas covered in this study, the Beaufort Sea featured the overall strongest average annual coastal erosion of 1.12 m. Several quality layers are provided for both the DL coastline product and the CVA-based coastal change analysis to assess the applicability and accuracy of the output products. The predicted coastal change rates show good agreement with findings published in previous literature. The proposed methods and data may act as a valuable tool for future analysis of permafrost loss and carbon emissions in Arctic coastal environments.

**Keywords:** permafrost; coastal erosion; circum-Arctic; deep learning; change vector analysis; Google Earth Engine; synthetic aperture RADAR

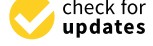



## 1. Introduction

Roughly one-quarter of the terrestrial area in the Northern Hemisphere and about two-thirds of exposed land north of 60° latitude are underlain by permafrost [1,2]. Moreover, stored carbon stocks within the frozen ground material are estimated to be twice as high compared to the amount of carbon that is currently present in the atmosphere [3,4]. However, increasing ground temperatures are reported for most regions underlain by permafrost [4–6]. As a consequence, future predictions on the distribution of frozen ground suggest a drastic reduction in the permafrost extent [7,8]. A thawing of permafrost leads hereby to the release of the stored organic materials via greenhouse gases, which may cause trillions of dollars in global economic damage without mitigation action [3,9].

A widespread phenomenon that is associated with the deteriorating state of frozen ground is the erosion of permafrost coastlines [10–12]. Several studies highlighted the accelerated erosion of Arctic coastlines in recent years [13,14]. It was further reported that average erosion rates more than doubled for unlithified coasts of Canada, Alaska, and Siberia since the beginning of the century [15]. Several drivers and their interplay

are responsible for the continuous retreat of Arctic coastlines. The thawing of frozen ground itself causes a destabilization of the coastline that results in increased erosion vulnerability [13,14]. In addition, environmental factors, such as the increase in storm frequency [16], rising sea and air temperatures [17–19], the decrease in sea ice extent [20–23], and the increased duration of the open-water period [24,25] all amplify erosion processes on permafrost coasts [13,14,26]. Moreover, a continuous sea level rise is predicted to strongly affect the cliff retreat until the end of the century [27,28]. Drastic changes in Arctic coastal environments can be observed as a consequence of increasing erosion rates of permafrost coasts. Fish and wildlife habitats are changed [29–31] and human settlements and infrastructure are at risk of damage [27,32–34]. In addition, previously stored carbon stocks are released into the oceans [29,34–38]. The release of stored carbon stocks from coastal erosion is hereby expected to rise by up to 75% until the end of the century [39]. Figure 1 illustrates an example site affected by coastal erosion within the permafrost domain along the coastline of the Tuktoyaktuk Peninsula in Canada.

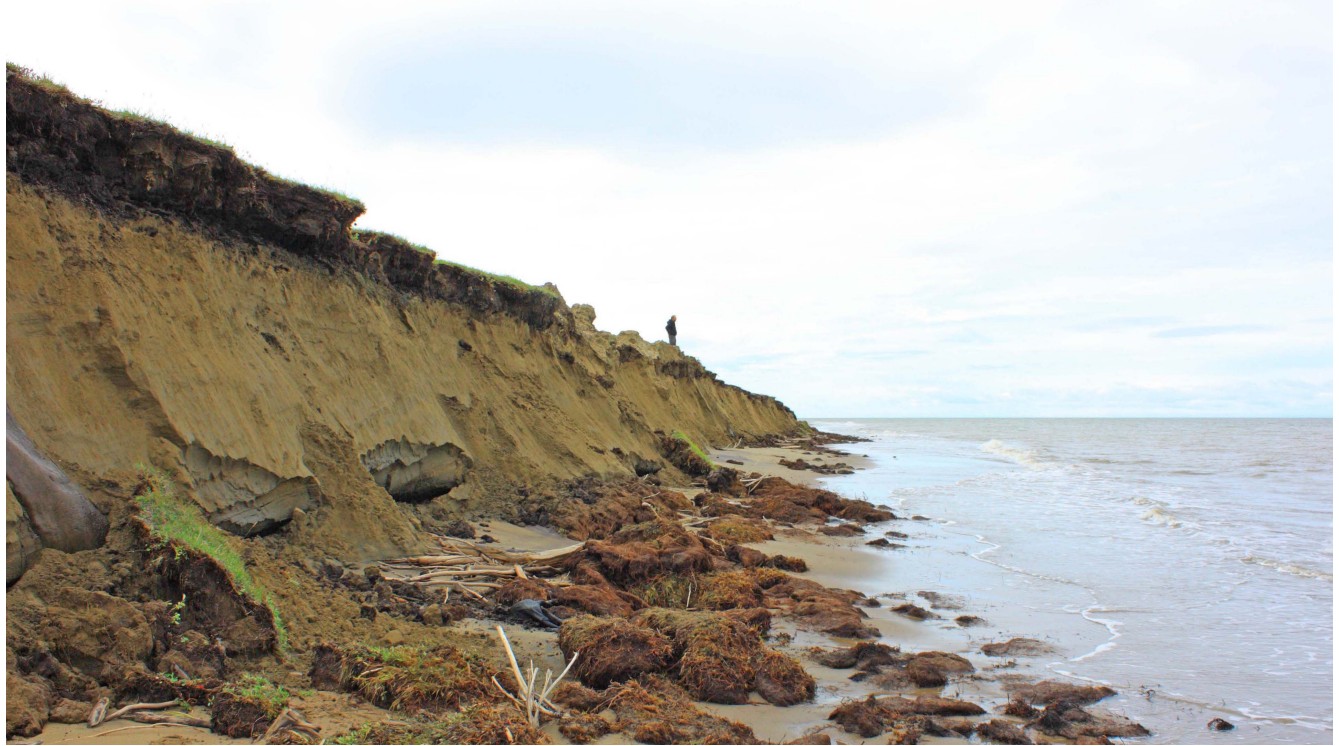

**Figure 1.** Coastal erosion along a permafrost coastline of the Tuktoyaktuk Peninsula in Canada. Photo taken in July 2012 by Tobias Ullmann.

This highlights the need for large-scale and high-resolution quantification of Arctic coastal change in order to fully assess the impacts of eroding permafrost coastlines not only on the environment, but also on human infrastructure and society. Satellite remote sensing is hereby a powerful tool for fast, inexpensive, and spatially explicit analysis over large spatial scales [40]. It is particularly valuable for analyzing remote and difficult-to-access regions. However, challenging environmental conditions in the form of low sun angles, frequent cloud cover, and low light intensities (including polar night) limit the usability of optical data in Arctic environments [41,42]. Synthetic Aperture RADAR (SAR) data on the other hand is largely independent of the aforementioned environmental conditions and serves therefore as a very attractive data source for studying these regions [43]. Bartsch et al. [44] investigated in a recent study the applicability of different SAR wavelengths in the context of detecting coastal erosion. Although the general application was rated to be feasible, issues with viewing geometries, inconsistencies in data acquisition, and ambiguous scattering behavior proved to be considerable challenges [44]. Another

recent study by Philipp et al. [40] demonstrated a significant reduction in noise within the backscatter signal by working on annual composites instead of single observations.

First efforts in circum-Arctic erosion quantification were made by Lantuit et al. [14], who provided a geomorphological classification for 1315 segments covering over 100,000 km of Arctic coastline in the form of the Arctic Coastal Dynamics Database (ACD) database. Next to the coastal change rates, the database also provides information about parameters, such as shore material, volumetric ground ice content, soil organic carbon content, and several others [14]. A recent study by Rolph et al. [45] presented the first physics-based model to simulate coastal retreat rates at the circum-Arctic scale, called "ArcticBeach v1.0". Simulated coastal erosion was thereby in the same order of magnitude as the observed erosion rates based on two test sites [45].

This study aims to build a comprehensive monitoring framework for current and future coastal erosion on a circum-Arctic scale and high resolution based on satellite remote sensing data. It therefore aims to close existing gaps by providing an inexpensive, robust, reproducible, and ongoing observation approach with high spatial resolution (10 m) on an annual basis. It hereby further represents the logical continuation of the recently published work by Philipp et al. [40] through the application of the same methods on a circum-Arctic scale. Thus, the goals of this study are (1) to generate a high quality and circum-Arctic coastline product using a Deep Learning (DL) workflow in combination with annual Sentinel-1 (S1) backscatter composites, and (2) to quantify pan-Arctic erosion and build-up rates with high spatial resolution via a Change Vector Analysis (CVA) approach. In addition to manually digitized reference data, the OpenStreetMap (OSM) coastline was used for training the DL networks. Although quality fluctuations across different regions are frequently reported for OSM [40,46], the vast amount of additional training data outweighs the variations in data quality for neural networks, which are reported to be comparatively error resistant [47,48].

## 2. Materials and Methods

The study was generally divided into three parts. First, the study area was selected and S1 Ground Range Detected (GRD) backscatter images in Interferometric Wide (IW) swath mode were pre-processed. Annual median and standard deviation (sd) backscatter composites were generated for the years 2017–2021. Annual composites represent the months June–September. The second step was dedicated to the generation of a DL-based high quality and circum-Arctic scale coastline product which acted as a basis for all further analysis. Nine different U-Net architectures have been combined to generate a DL-based coastline product covering a total of 161,600 km of the Arctic coastline. Lastly, coastal change was quantified in proximity to the DL reference coastline via a CVA approach in combination with the annual S1 backscatter composites. In addition to the DL coastline and the CVA-based coastal change quantification, several quality layers were provided to assess the applicability of the proposed data and methods across different regions, and for quality control of the final output products. In addition, the impact of tidal changes on the analysis was investigated. Changes in local tides may have a significant influence on the exact location of the transition zone between land and sea, especially for flat sandy coasts. Each processing step and product are described in detail in the following sections.

A variety of data from different sources were used throughout this study. Mainly, S1 GRD images in IW swath mode [49] were applied for the analysis. Imagery from Sentinel-2 (S2) [50] and Google Earth [51] acted as a means for additional quality control. In case of Google Earth, high-resolution imagery from Maxar Technologies and Centre national d'études spatiales (CNES)/Airbus were available. Furthermore, the Climate Change Initiative (CCI) permafrost fraction dataset by Obu et al. [52] as well as the Arctic coastline product derived from OSM [53] served to define the study area. Moreover, Mean Tidal Level (MTL) information by the National Oceanic and Atmospheric Administration (NOAA) [54] from four buoy stations were incorporated to study the impact of tidal changes on the proposed methods and data. Further quality control was achieved through the

Global Land Ice Measurements from Space (GLIMS) glacier database [55] and information about daily sea ice concentration via the ARTIST Sea Ice (ASI) dataset [56]. Lastly, the coastal erosion rates based on the proposed data and methods in this study were compared to coastal change information provided in the ACD by Lantuit et al. [14]. Details on the applied data, such as the data type, spatial and temporal resolution, as well as temporal coverage within this study are listed in Table 1.

**Table 1.** List of data utilized within the framework of this study. The column "Temporal Coverage & Resolution" provides information about the temporal window of used data in this study, as well as the frequency of available data within this time span in parentheses.

| Name | Data Type | Spatial Resolution | Temporal Coverage & Resolution | Reference |
|---|---|---|---|---|
| Sentinel-1 Ground Range Detected (GRD) Interferometric Wide (IW) swath | Raster | 10 m | 2017–2021 (up to 6 days) | [49] |
| Sentinel-2 | Raster | 10 m | 2017–2021 (up to 5 days) | [50] |
| Google Earth | Raster | varies | 2017–2021 (varies) | [51] |
| Climate Change Initiative (CCI) Permafrost Fraction | Raster | 927 m | 2017 | [52] |
| OpenStreetMap (OSM) | Vector | - | 2022 | [53] |
| Buoy Mean Tidal Level (MTL) Data | Table | - | 2020 (6 min) | [54] |
| Global Land Ice Measurements from Space (GLIMS) glacier database | Vector | - | 2022 | [55] |
| ARTIST Sea Ice (ASI) Arctic Sea Ice Concentration | Raster | 3125 m | 2017–2021 (daily) | [56] |
| Arctic Coastal Dynamics Database (ACD) Database | Vector | - | 2012 | [14] |
| International Hydrographic Organization (IHO) Sea Areas | Vector | - | 2018 | [57] |

*2.1. Study Area*

The first major part of this study was dedicated to the selection of the study area. It is limited to Arctic coastal areas in proximity to permafrost occurrences and with available S1 GRD imagery in IW swath mode. The Arctic coastline product from OSM was used as the basis as it proved to have the highest overall accuracy compared to other publicly available coastline products across 10 test regions in the Arctic [40].

As a first step, the CCI permafrost fraction for the year 2017 by Obu et al. [52] was used to define the extent of permafrost. The data were converted into a binary map where pixel values are either 0 (no permafrost) or 1 ($\geq$1% permafrost occurrence). In order to also include smaller islands in proximity to the coastline that are not covered by the dataset, a buffer of 20 km was computed around the binary map.

The next step was to assess the spatio-temporal availability of S1 satellite imagery. For this purpose, the number of all S1 GRD imagery in IW swath mode for the months June–September and until the end of the year 2021 from 30 degrees latitude upwards was computed on a pixel basis (Figure 2a). Data access and filtering were conducted in Google Earth Engine (GEE). We further differentiated between imagery with an ascending or descending orbit. Based on the data frequency per orbit, a categorical map was defined that shows which orbit features the highest amount of S1 scenes per pixel (Figure 2c). This map served as a basis for any further processing of S1 data to make sure the most frequent orbit was used when filtering the data. If both orbits feature the same amount of images, S1 data were filtered to the ascending orbit. Moreover, the year of first available data per pixel is portrayed in Figure 2b. A binary for areas with one or more images present was created.

As a last step, the OSM Arctic coastline product was clipped to the binary maps of permafrost occurrence and S1 data availability to reveal the investigated coastline (Figure 3). A buffer of 10 km was applied on the clipped coastline in order to account for any potential inaccuracies of the OSM product.

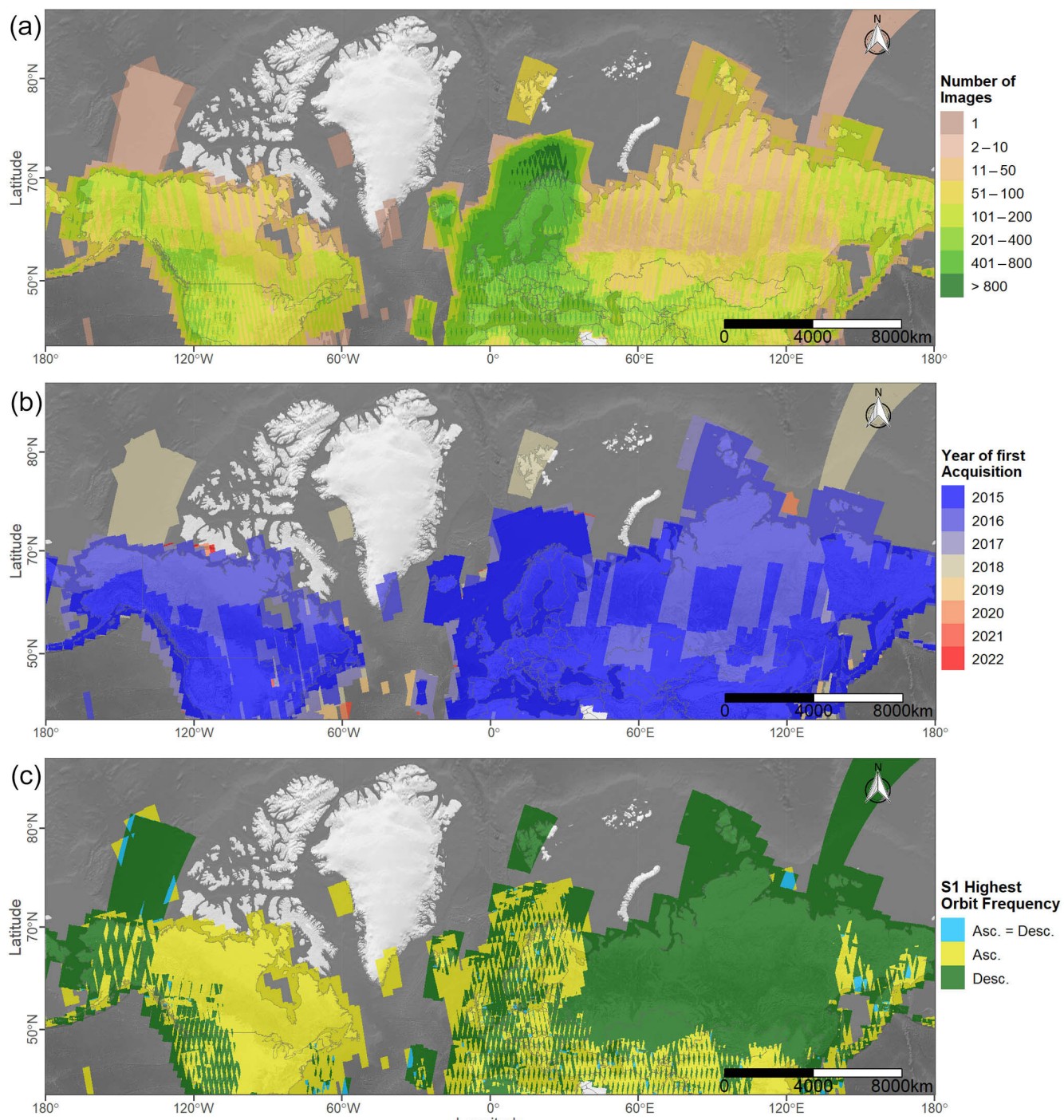

**Figure 2.** (**a**) Number of available Sentinel-1 (S1) Ground Range Detected (GRD) scenes in Interferometric Wide (IW) swath mode since launch and until the end of 2021 for the months June–September; (**b**) Year of first available S1 GRD scene in IW swath mode for the months June–September; (**c**) Orbit with the highest number of available S1 GRD scenes in IW swath mode since launch and until the end of 2021 for the months June–September. A shaded relief by Natural Earth [58] was utilized as a background map. Source of Administrative boundaries: The Global Administrative Unit Layers (GAUL) dataset, implemented by Food and Agriculture Organization of the United Nations (FAO) within the CountrySTAT and Agricultural Market Information System (AMIS) projects.

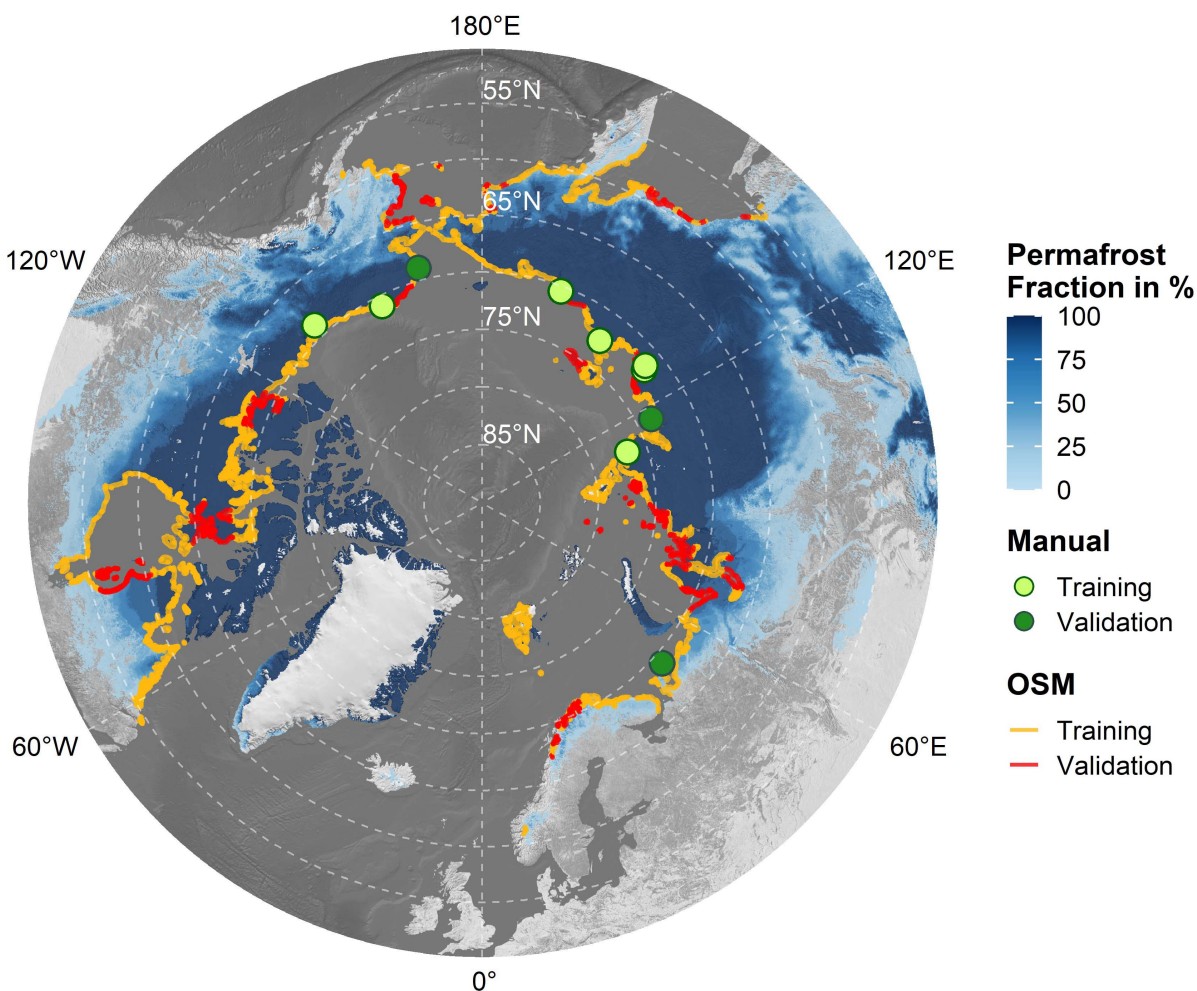

**Figure 3.** The study area divided into training and validation areas from the manually digitized sites (green points), as well as training and validation areas based on OpenStreetMap (OSM) (red and orange lines). The permafrost fraction across the Northern Hemisphere for the year 2017 based on data by Obu et al. [52] in combination with a shaded relief by Natural Earth [58] was utilized as a background map. All data is visualized in a polar Lambert azimuthal equal area projection.

### 2.2. Framework for Deep Learning-Based Arctic Coastline Extraction

The application of DL Convolutional Neural Networks (CNNs) has become increasingly popular in recent years and several studies testify to its capabilities, especially in the context of land vs. water segmentation [59–62]. One major objective of this study is to take advantage of the DL segmentation capabilities in order to generate a high-quality and circum-Arctic coastline product. The feasibility of combining S1 SAR data with a U-Net-based segmentation algorithm was already demonstrated by Philipp et al. [40]. In this study, the same concept is applied on a circum-Arctic scale with additional training via OSM as reference data. A detailed overview of how a CNN-based U-Net structure works is given by the previous publication by Philipp et al. [40] and in the original paper by Ronneberger et al. [63].

#### 2.2.1. Preparation of Sentinel-1 Pseudo-RGB Images

SAR data were derived from S1 features continuous observation capabilities due to its nature of being largely independent of sun illumination and weather conditions [43,64]. For that reason, S1 Level-1 GRD backscatter imagery in IW swath mode was employed for this study. The satellite data is available as backscatter coefficient sigma nought ($\sigma^0$) in the unit decibel (dB) with a spatial resolution of 10 m and was accessed via the cloud

computing platform GEE [65]. Imagery for the year 2020 and for the months beginning of June until the end of September was selected. Scenes between June and September were chosen to reduce the influence of sea ice contamination. The data were filtered to the most frequent orbit for a given region based on the previous data availability analysis (Figure 2c). The amount of speckle in each scene was reduced by applying a 3 × 3 median Moving Window (MV). Images were temporarily converted from dB to natural for the removal of speckle. As a next step, median and sd backscatter images for each polarization, vertical-horizontal (VH) and vertical-vertical (VV), were computed. While the median backscatter tends to be generally lower over water compared to land, the opposite effect can be observed for the sd backscatter. Here, the sd backscatter proved to be higher over water and lower over land [40]. This information was utilized to create Pseudo-Red Green Blue (RGB) images, with the median backscatter in VH polarization as the red channel, median backscatter in VV polarization as the green channel, and sd backscatter in VV polarization as the blue channel. Each scene was further normalized to their 2nd and 98th percentile. Figure 4 visualizes each of the channels, as well as the combined Pseudo-RGB for an example region in Shoalwater Bay, Canada. These Pseudo-RGB images served subsequently as inputs for training the U-Net models. In addition to the Pseudo-RGB images, the number of available scenes on a pixel basis was extracted as a quality layer for the final DL coastline product. All data access, filtering, and pre-processing were conducted in GEE.

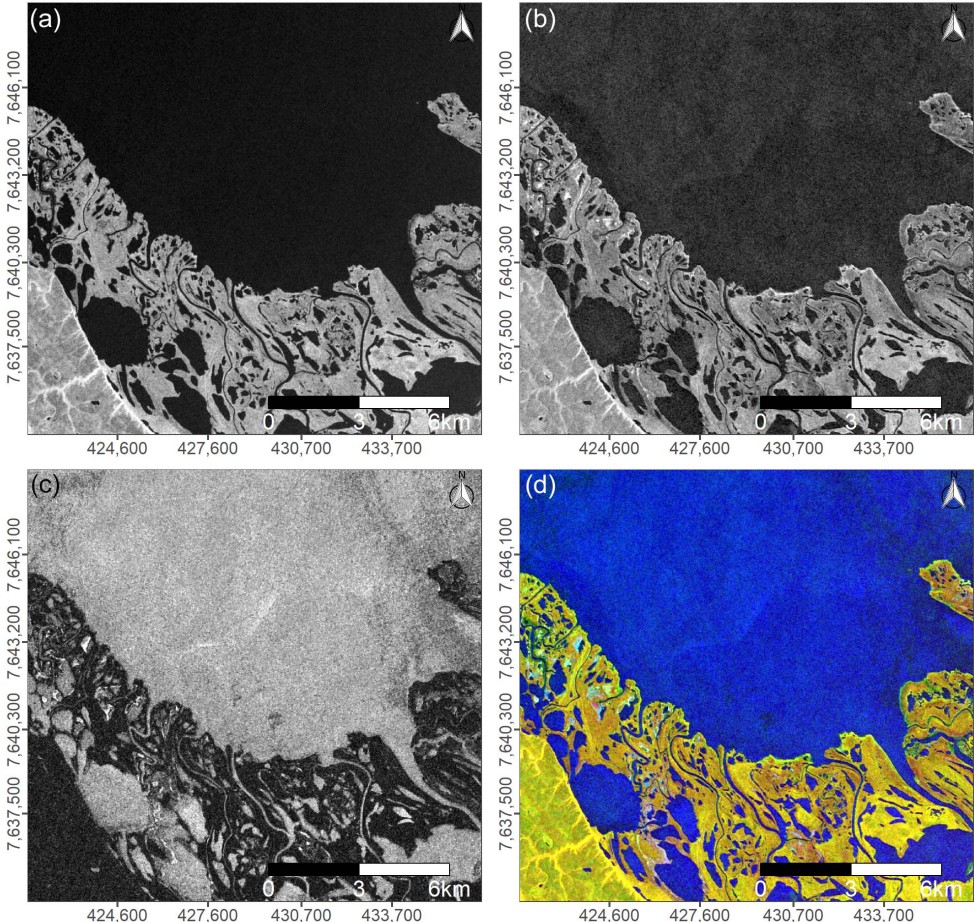

**Figure 4.** Section of the Shoalwater Bay region in Canada visualized by (**a**) annual median vertical-horizontal (VH) backscatter; (**b**) annual median vertical-vertical (VV) backscatter; (**c**) annual standard deviation (sd) VV backscatter; and (**d**) a pseudo Red Green Blue (RGB) composite of (**a–c**) based on Sentinel-1 (S1) Ground Range Detected (GRD) scenes from the beginning of June until the end of September 2020.

### 2.2.2. Preparation of Training and Validation Data

Training and validation data were collected on two levels. In the first level, a total of 1038 km of Arctic coastline and a combined area of 19,275 km$^2$ split into ten different regions across the Arctic were manually digitized. Details on the digitization process and the individual regions can be found in Philipp et al. [40]. Each region was associated with significant erosion rates based on the ACD by Lantuit et al. [14] and was therefore selected as a study area. The final data in the form of binary images (1 = land; 0 = water) was split into seven regions for training and three independent regions for validation. Therefore, spatial auto-correlation between the train and test dataset could be avoided.

The second level was dedicated to generating further training and validation data based on OSM. In order to cover the variety of different coastal morphologies and to overcome the limitation of CNNs requiring a vast amount of training data, OSM was utilized as an additional reference source. OSM is one of the biggest Volunteered Geographic Information (VGI) projects and currently features over eight million contributors OpenStreetMap [66]. Data in OSM is derived from multiple different sources and imported and edited by various editors [46]. The quality of imported data strongly depends on the source of the created geometries, such as aerial images or Global Positioning System (GPS) traces [46,67]. Multiple companies including Microsoft Bing, Yahoo!, and Aerowest provided (temporal) access to their aerial image database for the OSM project [46]. Although the overall quality of the OSM proved to outperform other freely available Arctic coastline datasets, the accuracy of the data also varies across different regions [40]. Furthermore, although being updated regularly, the dataset may not accurately depict the current state of highly dynamic regions, e.g., changing Arctic coastlines. Having that said, the vast amount of additional data is assumed to outweigh the variations in data quality, especially when working with CNNs. For this purpose, OSM land polygons were downloaded for all areas within the study region that are not already covered by the manually digitized study areas. The polygons were subsequently converted to binary rasters where pixel values are 1 in the case of the terrestrial area and 0 in the case of the water area. The entire OSM reference dataset was split into 136 tiles, of which 27 tiles were randomly chosen for validation and the remaining 109 tiles for training. Figure 3 visualizes all training and validation data derived from manual digitization and from OSM. The previously generated Pseudo-RGB images in combination with the binary reference rasters from both the manual and OSM sites were subsequently used as inputs for training the U-Net models.

### 2.2.3. Deep Learning Coastline Detection

A total of nine different models were trained and their results were combined in order to perform a high-quality segmentation between sea and land areas (including inland rivers and lakes). Each of the following models ResNet34 [68], ResNet50 [68], VGG16 [69], VGG19, [69], Inception v3 [70], Inception-ResNet v2 [71], DenseNet121 [72], ResNeXt50 [73] and SE-ResNeXt50 [74] were available with pre-trained encoder weights based on the ImageNet database ($\approx$14 Mio. images).

Additional model training was conducted in two stages. First, reference data from the manually digitized sites were utilized to train each model. The input data were converted to tiles the size of 512 by 512 pixels. A total of seven augmentations were applied to the input data, resulting in 49,096 tiles (32,606 tiles for training; 16,490 tiles for validation). As for the hyper-parameters, a Root Mean Square Propagation (RMSprop) optimizer with a learning rate of 0.001, a batch size of 8, a binary cross-entropy loss function, and binary accuracy as an accuracy metric were used. Each model was trained for 30 epochs. The number of epochs describes how often the entire dataset is presented to the network for training. The epoch with the highest binary validation accuracy was hereby treated as the representative trained model.

Each of the representative models received further training in the second stage by using reference data from the OSM sites. In the case of the OSM reference data, no augmentation was applied. The number of 512 by 512 pixels tiles from OSM hereby totaled 307,056

(237,460 tiles for training; 67,760 tiles for validation). The same hyper-parameter settings were applied for the training with data from manually digitized sites. Again, the models were trained for 30 epochs, and the epoch with the highest binary validation accuracy acted as the representative output per architecture.

The fully trained models were subsequently utilized to create probability maps with values ranging between 0 and 1 for each 512 by 512 pixels tile across the entire area of study. A threshold of 0.5 was applied on each probability map to differentiate between land (including inland rivers and lakes) and sea area. As a result, nine binary maps, one from each model, were available per tile. In order to receive the most representative output class, the mode value was computed across all nine binary segmentation maps on a pixel basis. In addition to the mode, the (dis-)agreement between the models was assessed as a quality layer for the DL-based final coastline product. The formula for the normalized model agreement is shown in Equation (1). The value ranges between 0.11, if only 5 out of 9 models agreed on the output class, and 1, if the output of all models was the same class. Lastly, objects that are smaller $\approx$0.2 km$^2$ were removed, and holes that are smaller $\approx$3 km$^2$ were closed in the final binary classification map. The border between sea and land area (including inland rivers and lakes) was vectorized, revealing the DL-based coastline product. A final screening of the coastline product was conducted and minor local adjustments were manually applied.

$$model_{agreement} = \frac{n_{mode} - \frac{n_{models}}{n_{classes}}}{n_{models} - \frac{n_{models}}{n_{classes}}} \tag{1}$$

where:

$n_{mode}$ = Number of occurrences of the mode value;
$n_{models}$ = Total number of models;
$n_{classes}$ = Total number of classes.

### 2.3. Quantification of Coastal Change

As described in detail by Philipp et al. [40] and as shown in Figure 4, median backscatter and sd backscatter tend to show inverse behavior when comparing land and water areas. While the median backscatter is generally higher over land and lower over water, the opposite can be observed for the sd backscatter. This behavior can be exploited to analyze changes between land and water via a CVA approach. CVA is a commonly applied tool in change detection analyses and allows not only for the identification of the change direction, but also the magnitude of change [75]. Furthermore, CVA avoids an accumulation of errors from separate input classifications in contrast to traditional post-classification change detection approaches [76]. Therefore, CVA in combination with S1 backscatter was employed to quantify coastal change on a circum-Arctic scale.

#### 2.3.1. Magnitude of Change via Change Vector Analysis

In order to conduct the CVA, annual median and annual sd backscatter for the years 2017 and 2021 were computed. For areas where no images were available for the year 2017, scenes from 2018 were used instead. Similar to the pre-processing of Pseudo-RGB images, S1 GRD scenes in IW swath mode were filtered to both years and to the months June–September. Images were further filtered to the most common orbit per area based on the previously calculated orbit frequency map (Figure 2). Moreover, the speckle effect was reduced by applying a 3 × 3 median MV. Lastly, median and sd backscatter in VV polarization was computed for each year and used as inputs for the CVA computation. If the median backscatter decreased and the sd backscatter increased, it was interpreted as a change from land to water and thus, erosion. If a change in the opposite direction could be observed, it was interpreted as build-up. The final magnitude of change maps was normalized to have values ranging between 0 and 1. Moreover, the number of available

images per year and per pixel were assessed as an additional quality layer for the final CVA coastal change product.

2.3.2. Post-Processing of Change Vector Analysis

The previously computed DL coastline product served as a reference area for the coastal change analysis. A one-sided buffer with a size of 200 m was computed in the direction of the sea, while a 50 m buffer was added in the direction of land. The magnitude of change maps was clipped to the buffered coastline area in order to limit the change detection to the coastline. Next was the identification of suitable thresholds for differentiating between actual change and noise within the magnitude of change maps. For this purpose, manual delineation of coastal change was conducted for the manual test sites based on the analyzed S1 scenes in combination with optical data from S2 and high-resolution imagery from Google Earth. These manually digitized coastal changes served as a reference for identifying the most suitable threshold values (0.35 for erosion and 0.6 for build-up). Details on creating the reference data and extracting optimal threshold values can be found in Philipp et al. [40]. A 3 × 3 mode MV was applied on the thresholded change maps to reduce the amount of left-over noise to a minimum. Further processing included computing the distance between each erosion/build-up cluster and the DL reference coastline. Clusters of coastal change with a minimum distance larger than 100 m to the DL coastline were interpreted as noise within the water and therefore removed. Moreover, the effect of changing glaciers on the analysis was reduced to a minimum by removing areas that intersect with a 500 m buffer around any glacier polygons derived from the GLIMS glacier database [55,77]. Moreover, areas with less than ten available S1 GRD scenes in IW swath mode in either the year 2017 (or 2018) and/or 2021 were excluded from the analysis. Within the context of this study, it is assumed that a higher amount of available images generally leads to more robust change analyses. Therefore, and to avoid measuring noise instead of change, only areas with more than 10 scenes available for the months of June–September in both years were included. Furthermore, areas with less than 50% sea-ice-free days in either year during the observation period June–September were excluded from the analysis. A pixel was considered contaminated by sea ice if at least 20% ice was present based on daily sea ice concentration information from the ASI database [56]. Finally, average change rates in the form of erosion and build-up were computed separately for 400 m segments along the DL coastline. For each segment, a rectangular polygon the size of 40 × 40 pixels (400 × 400 m) around a center point on the coastline was generated. Next, the number of erosion/build-up pixels in this area was extracted. Lastly, the average change (either build-up or erosion) per segment could be calculated as shown in Equation (2).

$$change_{seg} = length_{window} * \frac{n_{change}}{n_{total}}$$

(2)

where:

$change_{seg}$ = Average change (either erosion or build-up) per segment in meters;
$length_{window}$ = Length of the rectangular observation window in meters;
$n_{change}$ = Number of pixels that indicate change (either erosion or build-up);
$n_{total}$ = Total number of pixels in the observed window.

*2.4. Validation and Quality Control*

Extensive efforts were made for validating both the DL coastline product and the CVA coastal change analysis. Next to accuracy statistics based on manually digitized reference coastlines, quality layers, such as the model agreement, the number of available images, and the the total number of days with sea-ice contamination during the observation period are provided on a pixel basis. Further analysis of the impact of tidal changes on identifying the coastline was performed.

### 2.4.1. Tidal Influence on the Accuracy of the SAR-Based Coastline

The exact delineation of the Arctic coastline may vary throughout the observation period June–September due to tidal changes. Therefore, it is reasonable to not use single observations, and instead combine several images across an observation period to receive a representative output. However, based on the number of available images on the current tide level at the acquisition times, the quality of this aggregated image may vary. In order to assess the impact on tidal changes, MTL from four buoy stations on a six-minute basis and covering the same temporal window as the Pseudo-RGB images (June–September 2020) provided by the [54] were downloaded. MTL data from the following stations were used: 9468333 Unalakleet, 9468756 Nome, 9491094 Red Dog Dock, and 9497645 Prudhoe Bay. Figure 5 illustrates the distribution of buoy stations across Alaska.

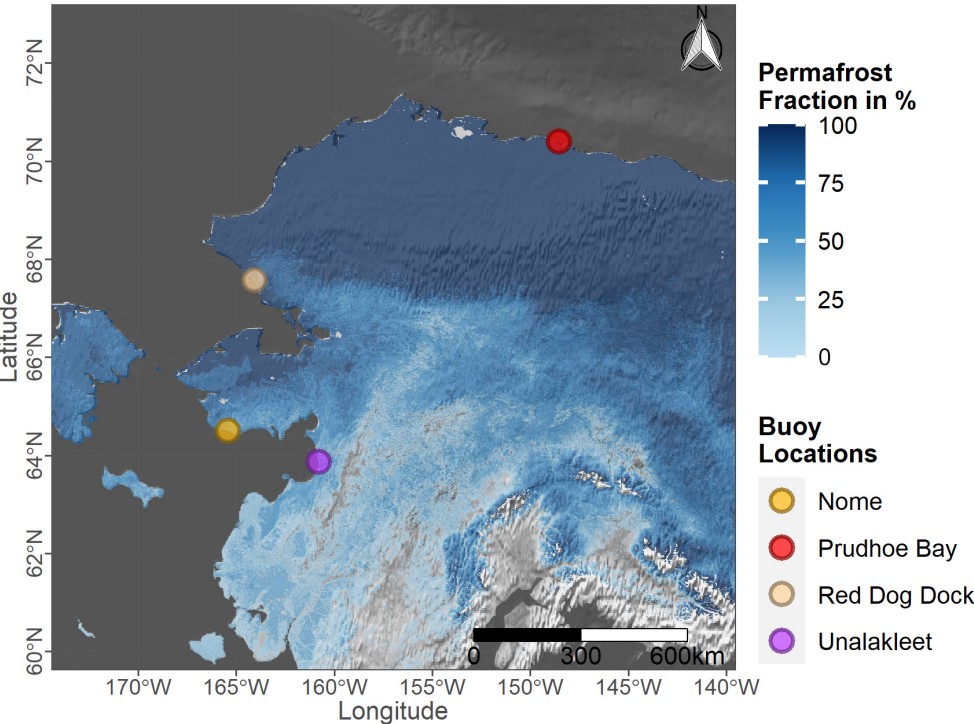

**Figure 5.** Locations of the buoy stations 9468333 Unalakleet, 9468756 Nome, 9491094 Red Dog Dock, and 9497645 Prudhoe Bay provided by the National Oceanic and Atmospheric Administration (NOAA) [54]. The permafrost fraction across the Northern Hemisphere for the year 2017 based on data by Obu et al. [52] in combination with a shaded relief by Natural Earth [58] was utilized as a background map.

The MTL describes the arithmetic mean of mean high water and mean low water [78]. The MTL for each acquisition date and time of available S1 GRD data in IW swath mode was extracted for each tide station and compared to the overall mean MTL across the entire observation period. It was ensured that both satellite data and buoy data are present in the same time zone (Greenwich Mean Time (GMT)). It is assumed, that a close mean MTL from S1 acquisition dates to the overall mean MTL across the entire observation period indicates a highly representative annual S1 composite.

### 2.4.2. Accuracy Assessment of Deep Learning Coastline

Statistics on the binary accuracy and loss values for each model are provided for the quality assessment of the individual model outputs. In addition, further accuracy assessment on the final combined binary classification map within a 500 m buffer around the coastline was conducted. Since high accuracy numbers can be expected for binary classifications across large regions, a focus was put on the transition zone between land

and sea area. For this purpose, common metrics, such as overall accuracy, precision, recall, and the $F_1$-score were derived. We differentiated between training and validation areas, as well as between manually digitized sites and OSM sites. The deviation between the 1038 km of manually digitized reference coastline and the generated final DL coastline product served as another means to quantify the quality of the predicted coastline. Moreover, several quality layers are available, such as the number of available S1 scenes, the level of agreement between the nine separate models, and the total number of days with more than 20% sea-ice contamination during the observation period via the ASI database [56].

### 2.4.3. Accuracy Assessment Coastal Change via Change Vector Analysis

As for the CVA analysis, manually digitized coastal changes across the manual test sites served as a basis for suitable threshold identification. Details on the creation of reference data for the CVA change maps are provided in Philipp et al. [40]. Moreover, the number of available S1 images, and the total number of days with sea-ice contamination based on the ASI database [56] are provided per year and per pixel as additional quality layers. The number of days with $\geq$20% sea ice for the time period June–September 2017 across the Arctic is visualized in Figure 6.

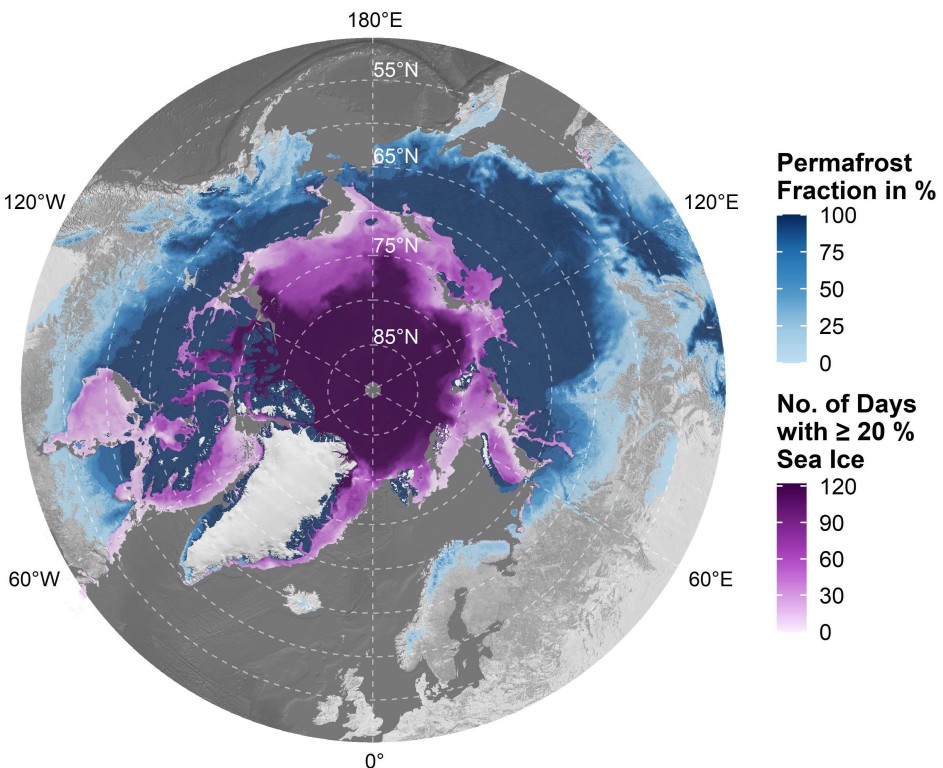

**Figure 6.** Total number of days with at least 20% sea ice between June and September 2017 based on the ARTIST Sea Ice (ASI) sea ice database [56]. The permafrost fraction across the Northern Hemisphere for the year 2017 based on data by Obu et al. [52] in combination with a shaded relief by Natural Earth [58] was utilized as a background map.

Lastly, information about the glacier extent using the GLIMS glacier database [77] was included for quality control. The magnitude of change maps themselves act as a further quality layer that can be used to identify custom threshold values based on image availability, sea ice contamination, and region.

## 3. Results

### 3.1. Deep Learning Coastline Detection

Accuracy and loss values after training exclusively with reference data from the manually digitized sites are already reported for each model in Philipp et al. [40]. The focus

of this study lies on the final accuracy statistics after further training with additional reference data generated through OSM. Thus, accuracy numbers for a significantly larger area are provided. Binary training accuracy for the final models after training on OSM sites ranged between 0.9838 (ResNet34) and 0.9952 (Inception v3). An average training accuracy of 0.991 could be observed. Validation statistics were similarly high and ranged between 0.9785 (Inception v3) and 0.9805 (ResNeXt50) with an average validation accuracy of 0.9792. Loss rates between 0.0121 (Inception v3) and 0.0488 (ResNet34) and an average loss of 0.0253 can be reported for the training sites. The minimum and maximum loss rates for the validation sites were 0.0696 (ResNet34) and 0.1508 (Inception v3), respectively. An average loss of 0.1051 was hereby observed for the validation areas. Details about the accuracy and loss values for each model in combination with the associated number of epochs are listed in Table A1 (Appendix A).

The final predicted coastline has a total length of roughly 161,600 km. Accuracy statistics within a 500 m buffer around the DL coastline product were overall slightly lower compared to the full scenes, but still relatively high across both training and validation areas sites. Overall accuracy for the manually digitized sites proved to be 0.95 in the case of training data and 0.97 in the case of validation data. Similarly, high numbers were observed for precision, recall, and the $F_1$-score for both terrestrial and sea areas and in both cases training and validation sites. As for the accuracy measures within the OSM sites, overall accuracy values of 0.95 for training areas and 0.94 for validation areas can be reported. Again, accuracy measures in the form of precision, recall, and $F_1$-score were revealed to show good agreement between the predicted binary segmentation maps and the reference data. Details about each accuracy measure for both training and validation sites and further separated into manually digitized areas and OSM areas are listed in Table 2.

**Table 2.** Accuracy statistics within a 500 m buffer around the manually digitized reference coastline (Manual) and OpenStreetMap (OSM) coastline for the final combined binary classification map after post-processing. Precision, Recall, and $F_1$-scores are given for both classes, terrestrial area (including inland lakes and rivers) and sea. Accuracy measures are rounded to the second decimal place.

| **Manually Digitized Sites** | | | | | |
|---|---|---|---|---|---|
| **Area** | **Overall Acc.** | **Label** | **Precision** | **Recall** | **F1** |
| Training | 0.95 | Terrestrial | 0.97 | 0.93 | 0.95 |
| | | Sea | 0.93 | 0.97 | 0.95 |
| Validation | 0.97 | Terrestrial | 0.98 | 0.96 | 0.97 |
| | | Sea | 0.96 | 0.98 | 0.97 |
| **OpenStreetMap (OSM) Sites** | | | | | |
| **Area** | **Overall Acc.** | **Label** | **Precision** | **Recall** | **F1** |
| Training | 0.95 | Terrestrial | 0.92 | 0.97 | 0.94 |
| | | Sea | 0.97 | 0.93 | 0.95 |
| Validation | 0.94 | Terrestrial | 0.90 | 0.99 | 0.94 |
| | | Sea | 0.99 | 0.91 | 0.95 |

The average deviation of the DL coastline product to the reference coastline is ±8.7 m in the case of the manually digitized sites and ±131.2 m in the case of the OSM sites. Median deviations of 6.3 m for the manual sites and 29.6 m for the OSM sites were measured. The sd values for manual sites and OSM sites were 8.5 m and 404.8 m, respectively. The minimum deviation was 0 m in both cases. In contrast, the maximum deviation across manual sites was 50 m, whereas the largest distance between the predicted line and the OSM reference was 8989.3 m. The 2nd and 98th percentile across manually digitized sites were 0.2 m and 36.9 m, respectively. For OSM sites, the 2nd and 98th percentile were 1.1 m and 1402.5 m.

Figure 7 illustrates an S1 annual Pseudo-RGB image, the associated binary classification in combination with the DL coastline product, as well as the level of agreement between individual classifications from each model for a section along Shoalwater Bay in Canada. The predicted coastline runs closely along the transition between sea and terrestrial area,

as expected from the generally high accuracy statistics. A high model agreement over the sea area can be observed. Model agreement varies across the terrestrial area. On the one hand, land area, small lakes, and small rivers are for the most part correctly attributed to the terrestrial class across all models. On the other hand, larger lakes and river deltas in particular cause confusion across different models. This confusion is partly also depicted in the binary classification as can be seen in the case of the larger river delta. The individual algorithms hereby disagree on the position of the border between the sea area and the start of the inland river, which belongs to the terrestrial class. As a result, the DL coastline product appears noisy in the transition zone of the river delta.

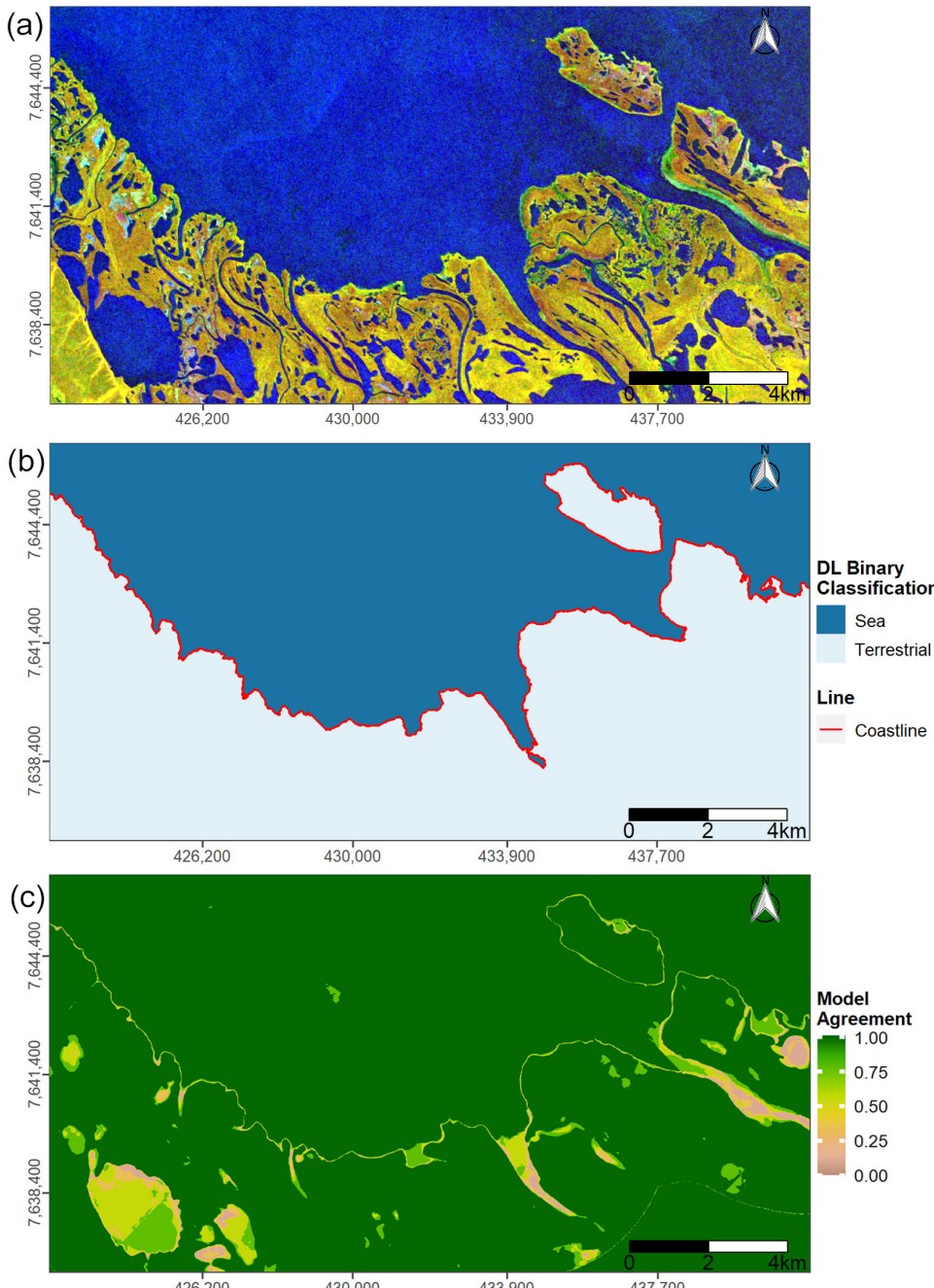

**Figure 7.** (**a**) Sentinel-1 (S1) pseudo Red Green Blue (RGB) composite covering the temporal window June–September 2020; (**b**) Mode image of the nine binary classifications from different U-Net architectures. The two classes represent terrestrial areas, including inland rivers and lakes (light-blue color) and sea areas (dark-blue color); (**c**) Agreement between the nine different classifications from different U-Net models. All images cover a section along Shoalwater Bay in Canada.

Another quality layer in the form of the number of available images per pixel is visualized in Figure 8. The Figure highlights varying S1 GRD IW swath data availability across the coastline of Alaska from June–September 2020. Up to 40 images were available for some regions across Alaska to generate the DL coastline product.

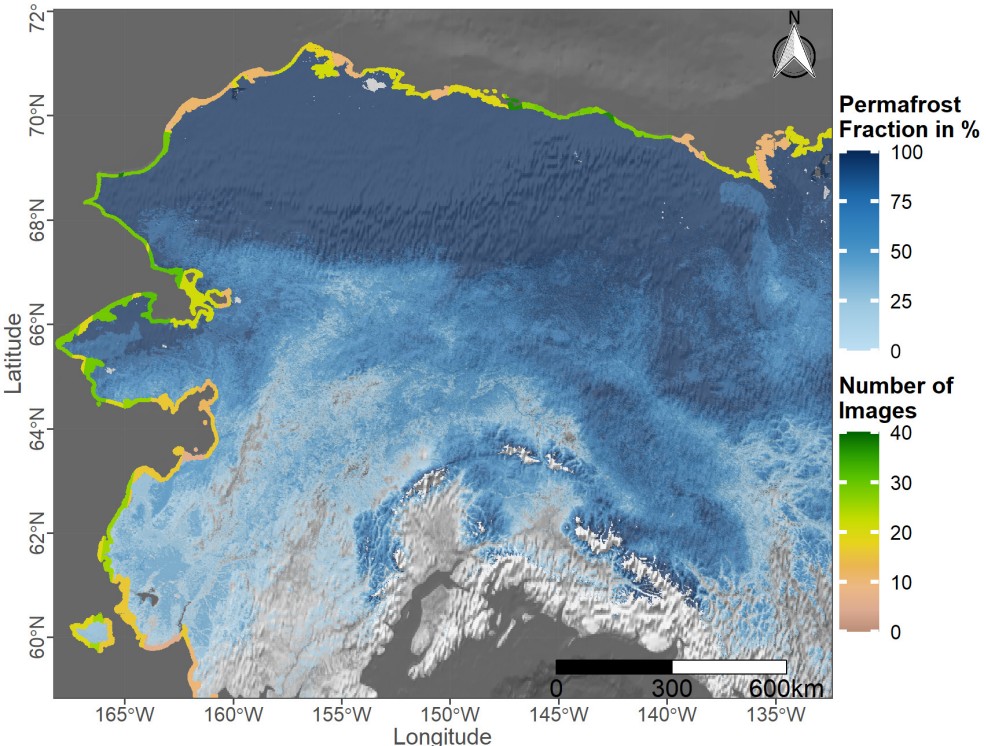

**Figure 8.** Number of available Sentinel-1 (S1) Ground Range Detected (GRD) scenes in Interferometric Wide (IW) swath mode over Alaska within June–September 2020 as a quality layer for generating the Deep Learning (DL) coastline product for the year 2020.

Figure 9 provides an overview of the Arctic coastal area covered by the DL coastline product. Due to a lack of available S1 GRD data in IW swath mode, the coverage of the Canadian Arctic Archipelago is strongly limited. Greenland was fully excluded from the analysis as no data were available over this region for the year 2020. The same applies to Severny Island, Yuzhny Island, and the Franz Josef Land archipelago in Russia. Figure 9b–e provide zoomed-in illustrations of the DL coastline product and a comparison to the OSM coastline. Figure 9b,c highlight the accuracy of the DL coastline product for the region Drew Point in Alaska.

OSM miss-classified coastlines and partly strong deviations can be observed in the reference background image, which represents the annual median backscatter in VV polarization from June–September 2020. Figure 9d,e, on the other hand, highlight an area where OSM outperforms the DL coastline product by including small islands that were removed during the post-processing in the DL product.

Figure 10 illustrates the changes in MTL across the temporal window June–September 2020 that was used for the generation of S1 Pseudo-RGB images. MTL data on a six-minute basis is visualized for the stations 9468333 Unalakleet (Figure 10a), 9468756 Nome (Figure 10b), 9491094 Red Dog Dock (Figure 10c), and 9497645 Prudhoe Bay (Figure 10d). The average MTL based on S1 acquisition dates for each respective region is highlighted by the red dashed line, whereas the overall average MTL across the entire buoy dataset per region is visualized by the grey dashed line. Although the MTL values feature strong fluctuations of up to 2.4 m across the observed time span and regions, both the average MTL from S1 acquisition dates and the overall average MTL from the entire buoy dataset were similar. Deviations of 0.23 m for Unalakleet, 0.09 m for Nome, 0.02 m for Red Dog

Dock, and 0.04 for Prudhoe Bay were observed. The number of available S1 scenes were 16 for Unalakleet, 20 for Nome, 39 for Red Dog Dock, and 22 for Prudhoe Bay. Therefore, a lower amount of available S1 data leads to a larger deviation to the average of the full buoy time-series.

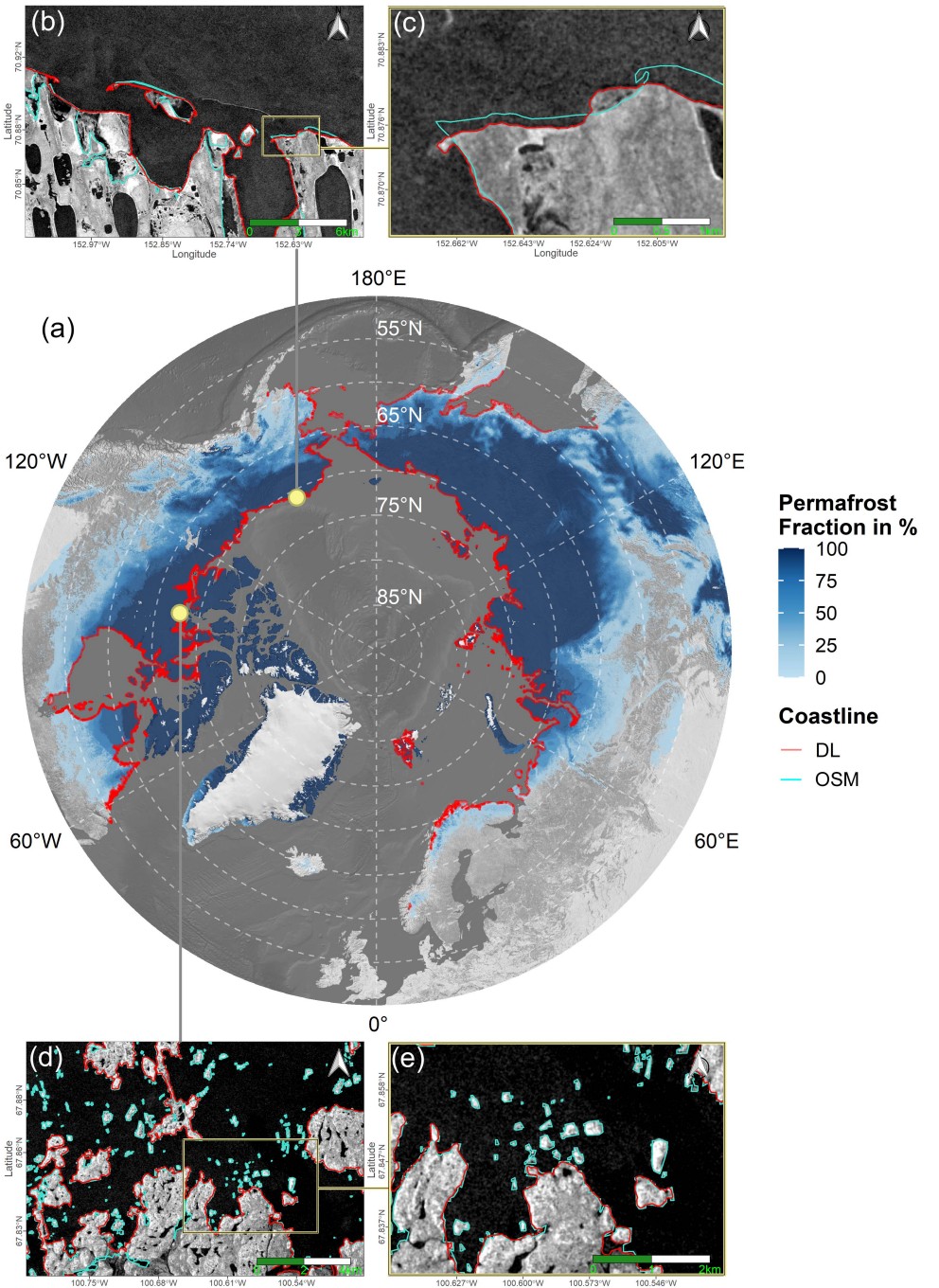

**Figure 9.** Circum-Arctic overview (**a**) and zoom-ins (**b–e**) on the Deep Learning (DL) coastline product (red line) and the OpenStreetMap (OSM) coastline (turquoise line) for the two different example regions Drew Point in Alaska (**b,c**), and an area in the Canadian Arctic Archipelago (**d,e**). The median Sentinel-1 (S1) backscatter in vertical-vertical (VV) polarization for the months June–September in 2020 was used as a background image for (**b–e**). The permafrost fraction across the Northern Hemisphere for the year 2017 based on data by Obu et al. [52] in combination with a shaded relief by Natural Earth [58] was utilized as a background map for (**a**).

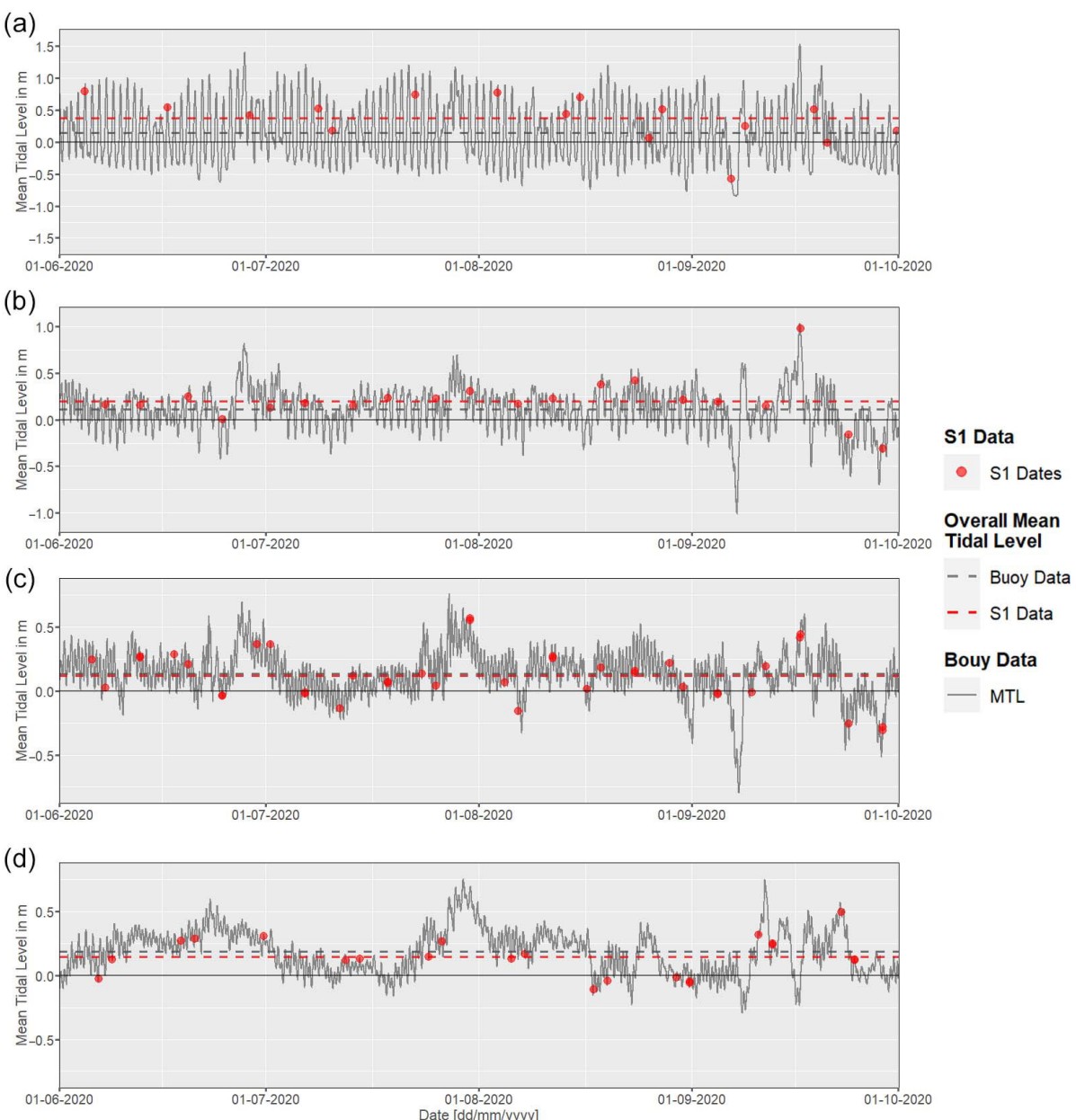

**Figure 10.** Mean Tidal Level (MTL) in m based on 6 min buoy data from June 1st to September 30th 2020 for the stations 9468333 Unalakleet (**a**), 9468756 Nome (**b**), 9491094 Red Dog Dock (**c**), and 9497645 Prudhoe Bay (**d**) provided by the National Oceanic and Atmospheric Administration (NOAA) [54]. The red points represent the MTL at Sentinel-1 (S1) acquisition times for the respective region. The dashed lines represent the overall MTL from June–September based on the full buoy dataset (grey line) and the tidal levels at the S1 acquisition times (red line).

## 3.2. Coastal Erosion and Build-Up Rates

Figure 11 visualizes CVA-based annual erosion both for the entire Arctic and zoom-ins on selected regions. The overview map (Figure 11a) shows average annual erosion rates for 20 km segments across the Arctic coastline for areas with more then 10 scenes available and less than 50% sea ice duration per year. Roughly 44.3% of segments feature no erosion. Around 35% show erosion rates between 0 and 1 m, and about 12.8% indicate erosion between 1 and 5 m. The remaining segments are made up of 3.8% with rates of 5–10 m, 2.4% with rates of 10–20 m, 1.3% with rates of 20–50 m, and less than 1% feature erosion rates of more than 50 m per year. Two zoom-in areas for segments with significant erosion rates are illustrated in the figure. Figure 11b–c feature a coastal area in Alaska.

The median backscatter in VV polarization for the months June–September is shown for 2017 (Figure 11b), 2021 (Figure 11c), and again the year 2021 together with the erosion-based land loss in red color (Figure 11d). The same principle applies to Figure 11e–g, but this time for a sandy delta in Russia. The algorithm categorizes changes in sandy deposits within the delta as coastal erosion.

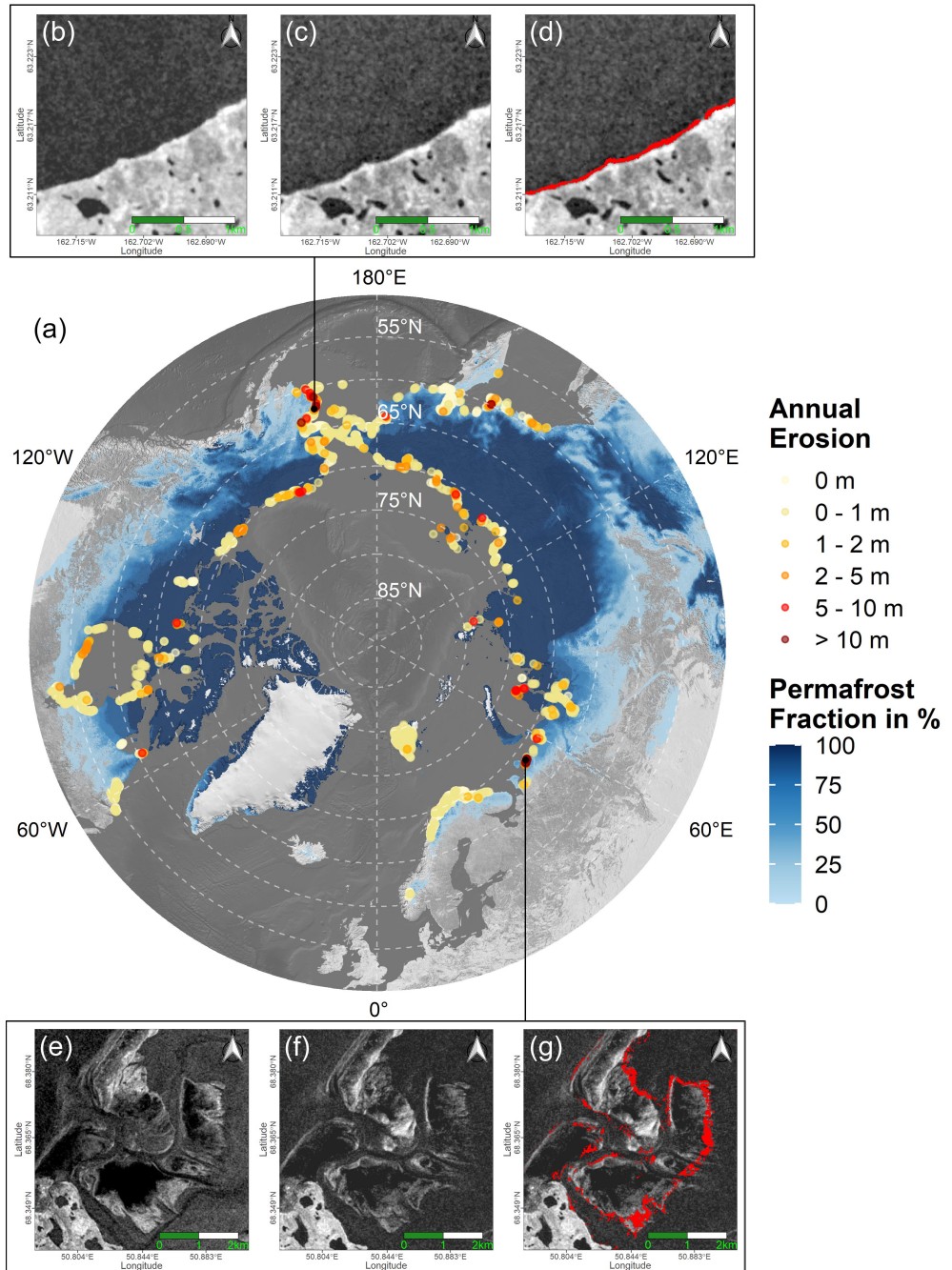

**Figure 11.** Circum-Arctic overview (**a**) and zoom-ins (**b**–**g**) on Change Vector Analysis (CVA)-based Arctic coastal erosion rates. The annual average erosion for 20 km segments is shown in (**a**). The median backscatter in vertical-vertical (VV) polarization from June–September in 2017 (**b**,**e**), 2021 (**c**,**f**), and again 2021 with the CVA-based erosion area (**d**,**g**) in red is shown for a coastal area in Alaska (**b**–**d**) and a sandy delta in Russia (**e**–**g**). The permafrost fraction across the Northern Hemisphere for the year 2017 based on data by Obu et al. [52] in combination with a shaded relief by Natural Earth [58] was utilized as a background map for (**a**).

Coastal change rates were investigated for different seas (Table 3). Annual statistics for a total of 16 seas based on the IHO Sea Areas dataset are available. The overall highest average annual erosion was observed for the Beaufort Sea (1.12 m), followed by the East Siberian Sea (0.91 m). The strongest average build-up can be reported for the Sea of Okhotsk (0.09 m), followed by the Laptev Sea (0.07 m). Overall maximum erosion is revealed for the Barents Sea (67 m) and the Bering Sea (62.5 m). Maximum build-up values were identified for the Laptev Sea (53.25 m) and the Barents Sea (52.67 m.). The highest 98th percentile values of erosion were extracted for the Bering Sea (9.75 m) and East Siberian Sea (9.5 m). The 98th percentile build-up values were mostly 0 m, with the exception of the Arctic Ocean (0.33 m), the Hudson Strait (0.25 m), and the East Siberian Sea (0.25 m). In terms of sd, the highest erosion sd values are associated with with the Barents Sea (3.63 m) and the Bering Sea (3.26 m). On the other hand, the strongest sd build-up values are present in the Laptev Sea (1.17 m) and the Barents Sea (0.97 m). Statistics in both Tables 3 and 4 are based on average erosion rates within 400 m segments along the coastline and therefore rounded to the second decimal place. However, when comparing coastal change on a per-pixel basis, the maximum accuracy can only be as high as the resolution of the used satellite data, in this case 10 m.

**Table 3.** Change Vector Analysis (CVA)-based annual erosion and build-up statistics per sea after the International Hydrographic Organization (IHO) sea areas and based on 400 m segments. The average, maximum, and the standard deviation (sd) are given. In addition, the percentage of segments with coastal change is provided. Build-up statistics are written in parentheses. Statistics are based on all segments, including segments with no coastal change.

| Sea | Mean | Max | SD | Perc. |
| --- | --- | --- | --- | --- |
| Bering Sea | 0.65 m (0.02 m) | 62.5 m (19 m) | 3.26 m (0.28 m) | 14.39% (1.13%) |
| Chukchi Sea | 0.19 m (0.01 m) | 26 m (11.25 m) | 1.06 m (0.21 m) | 10.02% (0.69%) |
| Beaufort Sea | 1.12 m (0.02 m) | 46 m (14.75 m) | 3.38 m (0.35 m) | 47.24% (1.32%) |
| Labrador Sea | 0.05 m (0 m) | 13 m (2.25 m) | 0.38 m (0.02 m) | 3.89% (0.07%) |
| Hudson Strait | 0.5 m (0.05 m) | 39 m (17.50 m) | 2.33 m (0.64 m) | 20.85% (2.07%) |
| Davis Strait | 0.73 m (0 m) | 38.75 m (0 m) | 3.03 m (0 m) | 18.92% (0%) |
| East Siberian Sea | 0.91 m (0.03 m) | 33.25 m (10 m) | 2.66 m (0.34 m) | 39.31% (2.66%) |
| Hudson Bay | 0.22 m (0.02 m) | 40 m (18.25 m) | 1.43 m (0.37 m) | 12.62% (1.22%) |
| The Northwestern Passages | 0.22 m (0 m) | 28.25 m (3.50 m) | 1.31 m (0.09 m) | 8.81% (0.30%) |
| Arctic Ocean | 0.05 m (0.01 m) | 3.67 m (1 m) | 0.31 m (0.1 m) | 3.72% (2.80%) |
| Barents Sea | 0.69 m (0.03 m) | 67 m (52.67 m) | 3.63 m (0.97 m) | 9.81% (0.91%) |
| Greenland Sea | 0.09 m (0.02 m) | 39.33 m (12.67 m) | 1.08 m (0.37 m) | 4.08% (1%) |
| Sea of Okhotsk | 0.56 m (0.09 m) | 43.75 m (23.75 m) | 2.89 m (0.93 m) | 12% (1.66%) |
| Kara Sea | 0.59 m (0.02 m) | 51.75 m (5.5 m) | 2.77 m (0.22 m) | 21.28% (1.92%) |
| Laptev Sea | 0.25 m (0.07 m) | 42 m (53.25 m) | 1.83 m (1.17 m) | 10.51% (1.67%) |
| Norwegian Sea | 0.01 m (0 m) | 18 m (5 m) | 0.26 m (0.08 m) | 0.75% (0.38%) |

Statistics on erosion and build-up rates per country are provided in Table 4. Annual CVA-based coastal change statistics for the five countries United States, Canada, Svalbard and Jan Mayen, Norway, and Russia are available. The strongest average erosion was observed for the United States (Alaska) (0.75 m), followed by Russia (0.62 m). The weakest average erosion could be observed for Norway (Scandinavian Peninsula) (0.01 m). The strongest average build-up values were observed for Svalbard and Jan Mayen (0.07 m) and the weakest, again, for Norway (0 m). Maximum annual erosion was observed for Russia (67 m), followed by the United States (Alaska) (62.5 m). Maximum build-up values are also attributed to Russia (53.25 m), followed by Svalbard and Jan Mayen (52.67 m). Both the 98th percentile and the sd of erosion were highest for the United States with 10.25 m and 3.45 m, respectively. The 98th percentile for build-up was 0 m across all countries. The highest sd build-up is shown for Svalbard and Jan Mayen (1.62 m). In total, 12.24% of segments indicated an average annual erosion rate of 3.8 m and a combined 17.83 km$^2$ of eroded land area per year, while 1.05% of segments featured an average annual build-p rate of 2.3 m and a combined annual build-up area of 1.02 km$^2$ across the entire investigated Arctic coastline. A total annual land loss of 17.84 km$^2$ and a total build-up area of 1.02 km$^2$

per year were observed across the entire covered coastline based on the proposed data and methods.

**Table 4.** Change Vector Analysis (CVA)-based annual erosion and build-up statistics per country and based on 400 m segments. The average, maximum, and the standard deviation (sd) are given. In addition, the percentage of segments with coastal change is provided. Build-up statistics are written in parentheses. Statistics are based on all segments, including segments with no coastal change.

| Country | Mean | Max | SD | Perc. |
|---|---|---|---|---|
| United States (Alaska) | 0.75 m (0.01 m) | 62.5 m (14.75 m) | 3.45 m (0.25 m) | 17.82% (1.05%) |
| Canada | 0.24 m (0.01 m) | 40 m (18.25 m) | 1.42 m (0.27 m) | 11.82% (0.67%) |
| Norway (Svalbard and Jan Mayen) | 0.09 m (0.07 m) | 39.33 m (52.67 m) | 1.01 m (1.62 m) | 4.06% (1.11%) |
| Norway (Scandinavian Peninsula) | 0.01 m (0 m) | 18 m (5 m) | 0.21 m (0.08 m) | 0.53% (0.32%) |
| Russia | 0.62 m (0.04 m) | 67 m (53.25 m) | 3.01 m (0.65 m) | 16.61% (1.68%) |

In addition, erosion rates derived from S1 and CVA were compared to coastal erosion estimates from the ACD. A total of 36.8% of ACD segments overlap with Arctic coastal areas investigated in the framework of this study. Therefore, comparisons are limited to those areas. CVA-based erosion and build-up numbers were combined and the average coastal change was computed for each ACD segment. Out of the remaining 484 ACD Arctic coastal segments, a total of 69.4% featured annual erosion rates that deviate less than 0.5 m from the CVA-based coastal change. Another 13% of segments featured differences of 0.5–1 m, 9.9% of segments showed differences between 1 and 2 m, and 7.6% exceeded 2 m in the difference of annual coastal change.

## 4. Discussion

### 4.1. A Deep Learning-Based Circum-Arctic Coastline Product

In this study, a novel circum-Arctic monitoring framework for quantifying annual erosion rates of permafrost coasts was presented. C-Band SAR data in the form of S1 GRD backscatter images in IW swath mode was combined with DL CNNs and CVA to generate a high-quality Arctic coastline product and quantify coastal change with high resolution and on an annual basis. Working with annual (June–September) S1 composites lowered the amount of speckle, reduced geolocation uncertainty of individual satellite scenes [79,80], and further limited the influence of sea ice contamination [14]. Annual S1 composites comprise both median and sd backscatter information. As seen in Figure 4, median backscatter was generally higher over land and lower over water, whereas the sd of backscatter intensity behaved inversely to the median. The higher sd of backscatter over water can be associated with frequent changes of the sea surface texture due to wind-driven capillary waves and gravity waves [81,82].

This backscatter behavior could be exploited for the generation of a DL-based Arctic coastline product by using the annual S1-RGB composites as inputs for nine different Convolutional Neural Networks (CNN) U-Net architectures. A major limitation of CNN-based algorithms is the need for vast amounts of training data [83]. In order to overcome this limitation, pre-trained Networks based on the ImageNet database (14 Mio. images) were used. Furthermore, augmentation was applied on the manually digitized reference areas covering 1038 km of coastline separated into 10 different Arctic regions. Thus, the amount of reference data was artificially increased seven-fold. Moreover, additional training and testing data were generated by using the OSM Arctic coastline product. Although the quality of the OSM data varies across different Arctic regions as reported in Philipp et al. [40], the additional amount of reference data outweighed the fluctuations in data quality. Neural networks are reported to be more error resistant compared to linear regression models [47]. An accuracy of 90% using a CNN could be achieved even though roughly one-third of the training data was erroneous [48]. Klein and Rossin [84] observed even a slight performance increase by introducing moderate amounts of errors (5–15%) in the training data to their Neural Network model [84]. Another limitation is identifying

the most suitable network architecture and depth for a given task [85]. Instead of relying on a single model, the results of nine different model architectures were combined in this study to generate the most representative output across a variety of different network types. All models produced training accuracies $\geq 0.9838$ and validation accuracies $\geq 0.9785$ (Table A1). The segmentation between the sea and the terrestrial area was therefore largely successful across all models. Statistics within a 500 m buffer around the reference coastline also reveal high accuracy values for the final combined binary segmentation map (Table 2). Both training and validation sites feature overall accuracy values of $\geq 0.944$. The final DL coastline product had a median deviation to the reference coastline of $\pm 6.3$ in the case of the manually digitized sites, and a deviation of $\pm 29.6$ m in the case of OSM sites. The higher deviation of the predicted coastline to the OSM line can be explained by the greater variety of coastal areas covered by the OSM reference data and the previously mentioned fluctuations in OSM data quality. Comparisons between the DL and OSM coastline products are visualized in Figure 9. On the one hand, DL outperforms the OSM coastline in the case of some areas by providing more accurate and up-to-date information. On the other hand, due to limitations of available S1 imagery for generating the DL coastline product, data coverage of OSM is significantly better. Moreover, smaller islands ($<0.2$ km$^2$) were removed during post-processing of the DL coastline, but are often present in the OSM dataset. Adjusting the minimum threshold for the object removal could lead to fewer excluded islands, but at the risk of introducing higher noise levels.

Investigations on the impact of tidal changes revealed good agreement between the average MTL for S1 acquisition dates and the actual average MTL within the observed time span based on data from four buoy stations (Figure 10). The higher the number of available S1 scenes for a given region, the lower the deviation of the average MTL for S1 acquisition dates to the actual average MTL. Therefore, the number of available satellite scenes may have a noteworthy influence on the accuracy of the average coastline position in the S1 annual composite images. That said, spatially more distributed and an overall higher amount of buoy data across the entire Arctic is needed to provide profound statements on the effect of tidal changes on Arctic coastlines. This applies especially to flat sandy coasts, where tidal changes can have a significant impact on the position of the border between sea and land. The number of available images per pixel represents therefore a valuable quality layer in these areas. Furthermore, computing median and sd composites for flat sandy coasts may lead to images with soft and difficult-to-distinguish transition zones between sea area and terrestrial area, which can have a negative impact on the accuracy of the predicted coastline.

### 4.2. Quantifying Coastal Change via Change Vector Analysis

The contrasting behavior between the median and sd backscatter for sea and terrestrial areas was further exploited in the CVA on coastal change. Maximum average annual erosion per country of 0.75 m (United States), and overall maximum erosion rates of up to 62.5 m (Russia) based on 400 m segments were revealed. The overall weakest erosion and build-up was observed in Norway. This can be attributed to the mostly lithified coasts that are less prone to erosion in this region [14]. The strongest average annual build-up (0.07 m) was reported for Svalbard and Jan Mayen, which may be explained by movements of small remaining glaciers that were not covered by the GLIMS database. Other build-ups can also be attributed to the accumulation of sandy deposits near coastlines and river deltas. The extracted erosion rates show good agreement with the findings of previous literature. Strong annual erosion of 20–50 m along Drew Point were observed via the CVA analysis. This matches with numbers published by Jones et al. [29] and Wang et al. [86], who reported erosion numbers for subsets of this area ranging between 6.7 and 22.6 m, and 30.8 and 51.4 m per year, respectively. Obu et al. [12] observed maximum coastal retreat rates of 10–17 m between 2012 and 2013 for the Bell Bluff site on Herschel Island in Canada, which agrees with findings of annual change based on the proposed CVA approach (12.5–15 m per year). The authors further communicated little to no (0–1 m)

coastal erosion along Kay Point near Herschel Island, which matches numbers from this study (0 m) [12]. Irrgang et al. [87] reported an average annual erosion rate of $0.7 \pm 0.2$ m for a 210 km long section of the Yukon coast in Canada on the basis of historical aerial and high-resolution satellite imagery from the 1950s until 2011. The proposed data and methods from this study suggest an average annual erosion of 0.5 m for the same area, which is in the equivalent order of magnitude as observed by Irrgang et al. [87]. Another study by [10] reported relatively small annual erosion rates of less than 1 m along the west coast of the Buor Khaya Peninsula in Russia, which also agrees with the findings in this study. In addition, over two-thirds of overlapping segments between CVA-based coastal change of this study and the ACD database feature coastal change rates with less than 0.5 m deviation, and therefore suggesting an overall good level of agreement between the two datasets. The strongest coastal change per sea was quantified for the Beaufort Sea in both the CVA coastal change analysis and the ACD database. Differences between the observed coastal change rates in this study and the ACD database can be attributed to differences in the observed temporal window, the applied spatial resolution, and uncertainties in both datasets.

### 4.3. Limitations and Future Potentials

The proposed methods and data provide a valuable tool for quantifying coastal erosion and build-up rates. However, the quality of the analysis heavily depends on the amount of available satellite data, which varies over time and space [88,89]. Figure 2 visualizes the varying data availability for different regions across the Arctic. Large parts of Russia and some regions in Canada feature relatively small amounts of available data, whereas the data frequency over Europe is relatively high. As of the time of writing this article, data is exclusively generated by S1A due to an on-board anomaly of the S1B satellite since 23 December 2021 [90]. Until the launch of S1C, limited data availability might have negative effects on the continuous analysis of the proposed monitoring framework. Quality layers in the form of the number of available images (Figure 8), number of sea ice days (Figure 6), model agreement (Figure 7), and the presence/absence of glaciers are provided on a pixel basis and may act as helpful proxies for assessing the applicability of the proposed methods and data, and the quality of the output products. The application of CVA in combination with S1 backscatter imagery was limited to areas with less than 50% sea ice days and more than 10 observations within the observed time span from June–September in order to minimize noise and miss-classifications. It also has to be mentioned that coastal change can only be meaningfully quantified, if the present erosion or build-up is larger than the resolution of the applied satellite data, in this case 10 m. The applied threshold values for CVA analysis, as proposed in Philipp et al. [40], led to best results for the tested areas, but optimal threshold values may vary depending on the coastal region, amount of available satellite data, and sea ice contamination. Especially for areas with low amounts of images and high sea-ice concentration, adjustment of the threshold values may be necessary to achieve the best results. In this context, the magnitude of change information acts as a valuable tool for differentiating between noise and actual change. Future data will allow for longer time series investigations. In this regard, the temporal window for the creation of composites could be extended over 2 years for areas with poor data availability. However, using a longer observation period for the creation of a single composite also increases the variation within this composite. Thus, there is a trade-off between having more data and having more uncertainty due to stronger changes across individual scenes used to generate a single composite image. Furthermore, by intersecting the extracted coastal change rates with additional information about the geomorphological parameters, such as lithification stage and ground ice content of Arctic coasts, as provided in the ACD by Lantuit et al. [14], may help to further understand varying erosion rates of permafrost coasts. Lastly, the long-term effects of the Fennoscandian land uplift [91] and the influence of sea level rise [27] on the proposed data and methods, especially for flat sandy coasts, should be investigated in future analyses. Although the proposed methods and data were applied and validated

exclusively within the Arctic permafrost domain, there is a high potential for transferability of this coastal monitoring approach to different latitudes.

## 5. Conclusions

In this study, Sentinel-1 (S1) Ground Range Detected (GRD) backscatter images in Interferometric Wide (IW) swath mode were combined with Deep Learning (DL) and Change Vector Analysis (CVA) in order to investigate erosion and build-up of permafrost coasts on a circum-Arctic scale. Annual median and standard deviation (sd) backscatter images were hereby utilized to generate a DL reference coastline and were further used as an input for CVA-based coastal change quantification. The following main conclusions can be drawn from this study:

- Despite fluctuations in data quality, OpenStreetMap (OSM) proved to be a feasible additional input for training Convolutional Neural Networks (CNN) U-Net architectures on the segmentation between sea and terrestrial areas in Arctic environments.
- DL in combination with annual median and sd backscatter from S1 allowed for the computation of a high-quality reference coastline with a total length of 161,600 km. A median accuracy of $\pm6.3$ m to the manually digitized reference coastline and a median agreement of $\pm29.6$ m to the OSM reference coastline was achieved.
- A good agreement between the average Mean Tidal Level (MTL) from S1 acquisition dates and the actual MTL was observed ($\pm0.02$–$0.23$ m). The higher the number of available S1 scenes, the smaller the gap between the MTL represented by the S1 acquisition dates and the actual average MTL for a given observation period.
- The inverse behavior of median and sd backscatter over sea and terrestrial areas could be successfully exploited for the CVA analysis. However, the quality and applicability of the analysis strongly depend on the number of available scenes, the present coast type, and total sea ice duration during the observed temporal window.
- Maximum annual erosion rates of up to 67 m were observed in Russia, followed by 62.5 m in Alaska. Overall average annual erosion was highest in the United States with 0.75 m, followed by Russia with 0.62 m. The weakest average annual erosion was observed in Norway (0.01 m). The Beaufort Sea featured the overall strongest annual average erosion of 1.12 m across all seas. Statistics are hereby based on all segments, including segments without coastal change.
- In total, 12.24% of the entire investigated Arctic coastline indicated an average annual erosion rate of 3.8 m and a combined 17.83 $km^2$ of eroded land area per year, while 1.05% of the coastline featured an average annual build-up rate of 2.3 m and a combined annual build-up area of 1.02 $km^2$.
- Quality layers in the form of the number of available images, number of sea ice days, model agreement, and the presence/absence of glaciers are provided on a pixel basis. The aforementioned quality layers may act as helpful proxies for assessing the applicability of the proposed methods and data, and the quality of the output products.

The proposed data and methods proved to be powerful tools for generating a high-quality Arctic coastline product, and for quantifying annual coastal change rates of Arctic permafrost coasts. The approach may also be feasible for different latitudes. The generated output products may further act as a means to quantify the loss of frozen ground, and for estimating carbon emissions in permafrost-affected coastal environments in future studies. The final circum-Arctic DL coastline product, CVA-based magnitude of change maps, and the associated quality layers will be made freely and openly available via the Earth Observation Center (EOC) Geoservice of the German Aerospace Center (DLR).

**Author Contributions:** M.P. and C.K. conceptualized the study design. M.P. processed, analyzed, and visualized the data and wrote the original manuscript. T.U., A.D. and C.K. contributed to the study concept, the writing, and the editing of the manuscript. All authors have read and agreed to the published version of the manuscript.

**Funding:** This publication received no external funding.

**Data Availability Statement:** All data that support the findings of this study are available from the corresponding author upon reasonable request.

**Conflicts of Interest:** The authors declare no conflict of interest. The funders had no role in the design of the study; in the collection, analyses, or interpretation of data; in the writing of the manuscript; or in the decision to publish the results.

## Abbreviations

The following abbreviations are used in this manuscript:

| | |
|---|---|
| ACD | Arctic Coastal Dynamics Database |
| AMIS | Agricultural Market Information System |
| ASI | ARTIST Sea Ice |
| CCI | Climate Change Initiative |
| CNES | Centre national d'études spatiales |
| CNN | Convolutional Neural Network |
| CVA | Change Vector Analysis |
| dB | decibel |
| DL | Deep Learning |
| DLR | German Aerospace Center |
| EOC | Earth Observation Center |
| FAO | Food and Agriculture Organization of the United Nations |
| GAUL | Global Administrative Unit Layers |
| GEE | Google Earth Engine |
| GLIMS | Global Land Ice Measurements from Space |
| GMT | Greenwich Mean Time |
| GPS | Global Positioning System |
| GRD | Ground Range Detected |
| IHO | International Hydrographic Organization |
| IW | Interferometric Wide |
| MTL | Mean Tidal Level |
| MV | Moving Window |
| OSM | OpenStreetMap |
| RGB | Red Green Blue |
| RMSprop | Root Mean Squared Propagation |
| S1 | Sentinel-1 |
| S2 | Sentinel-2 |
| SAR | Synthetic Aperture RADAR |
| sd | standard deviation |
| VGI | Volunteered Geographic Information |
| VH | vertical-horizontal |
| VV | vertical-vertical |

## Appendix A

**Table A1.** Accuracy statistics and epochs of final segmentation maps per model after additional training with OpenStreetMap (OSM) data. The epoch with the highest validation accuracy was chosen as a representation for each model. Accuracy and loss values were rounded to the fourth decimal place.

| Model | Epoch | Training Acc. | Training Loss | Validation Acc. | Validation Loss |
|---|---|---|---|---|---|
| DenseNet121 | 18 | 0.9894 | 0.0311 | 0.9796 | 0.1003 |
| Inception-ResNet v | 20 | 0.9933 | 0.0175 | 0.9796 | 0.1160 |

**Table A1.** *Cont.*

| Model | Epoch | Training Acc. | Training Loss | Validation Acc. | Validation Loss |
|---|---|---|---|---|---|
| Inception v3 | 29 | 0.9952 | 0.0121 | 0.9785 | 0.1508 |
| ResNet34 | 4 | 0.9838 | 0.0488 | 0.9790 | 0.0696 |
| ResNet50 | 27 | 0.9900 | 0.0276 | 0.9787 | 0.1000 |
| ResNeXt50 | 23 | 0.9885 | 0.0323 | 0.9805 | 0.0844 |
| SE-ResNeXt50 | 20 | 0.9941 | 0.0155 | 0.9799 | 0.1080 |
| VGG16 | 29 | 0.9945 | 0.0143 | 0.9787 | 0.1283 |
| VGG19 | 12 | 0.9900 | 0.0289 | 0.9787 | 0.0881 |

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
