# Peer review of "A Circum-Arctic Monitoring Framework for Quantifying Annual Erosion Rates of Permafrost Coasts"

_remotesensing, doi:10.3390/rs15030818_

Round 1
Reviewer 1 Report
This is an interesting paper about an important topic, based on careful and sophisticated analysis. The method proposed here could have widespread applicability, and the main improvements to the paper that I would suggest have to do with how to make the method easier to use in the future for monitoring of global coastal change. I have made some comments in the text, the most important of which I will elaborate on below.
The use of Open Street Map as a source of training data is unexpected and could attract criticism. We find out only in the Discussion that the authors chose OSM for a very good reason: accuracy is not crucial but widespread coverage is. I would suggest making this point early in the paper, possibly in the Introduction. Also, I think readers would be comforted if the authors could provide some information about the source of the OSM coastline. It presumably was vectorized from some kind of raster composite and could vary in age, quality, and resolution. This would be a good place to emphasize that these problems don't matter a lot, because the training data need only be right most of the time, and some errors are not important.
I was confused by the summary statistics for the coastal erosion rates. It appears the rates of coastal retreat were not computed using a transect-based method, which would have been impractical here. I suppose more likely the authors divided the area of land lost by the length of each coastal segment to obtain an average rate of loss over the segment. In any case, the Methods should include some description of this method. Also, it is always a bit confusing to present statistics on erosion rates, because normally we want to summarize the rate of erosion only for segments that had erosion, and separately summarize the rate of build-up for the areas that had build-up. If we do this then we must also include the respective lengths of areas that were subject to erosion vs. buildup, to provide an overall picture. For example, one might write that 90% of the coastline was eroding with an average rate of 5 m/year, and 10% was accreting at an average rate of 2 m/year. The 98th percentile values were confusing also because it was not obvious what set of data was used to compute them. Because we are interested in land area lost and with the potential for carbon release, I would suggest also summarizing the area of land lost.
Finally, land vs. water is a relatively easy image classification problem, and it is probably not necessary going forward to run 9 different and highly sophisticated DL models to map the coast. In fact, the main difference between the models, judging from Fig. 8, was how they coped with inlets, estuaries, and inland lakes, not the differentiation of land vs. water. To simplify the process for the future, could the authors suggest a smaller subset of models that would suffice? They even might be able to suggest a GIS-based technique for removing lakes and narrow inlets from the data set, rather than relying on the "black box" of DL to remove these non-target areas. Indeed, it is a simple matter in a polygon GIS layer to simply remove all water bodies that are not connected to the global ocean (i.e. lakes) from the results, no need to rely on DL for this. Estuaries and narrow inlets are a more difficult problem, but it would be comforting to have a rule-based method for excluding these (e.g., all inlets narrower than 500 m) if an automated method could be devised.

Author Response
First, we would like to thank the reviewer for taking the time to review our manuscript. We also would like to thank him/her for the positive general evaluation of our manuscript and for considering it to be an interesting article with careful and sophisticated analysis. Thanks to the feedback of the reviewer, we were able to further improve the quality of our manuscript. Please find our detailed answers in the attached pdf.

Reviewer 2 Report
The manuscript presents a circum-Arctic coastline product using a Deep Learning workflow in combination with annual Sentinel-1 backscatter composites from 2017 to 2021 to quantify pan-Arctic erosion and build-up
rates with high spatial resolution based on Change Vector Analysis approach. The manuscript is well-written and well-structured with good and clearly presented figures and tables. Circum-Arctic scale and high quality data have a high relevance for the scientific community, as well as an important significance for coastal settlement risk assessment due to coastal erosion.
My general comments are:
1. Should be used more reference in introduction and discussion part.
2. Wrong location of names on figure 5. Should be corrected on figure and should be checked correct usage of names throughout the text.
3. The discussion better to separate by subsections.
Detailed comments and suggestions are listed below:
Figure 1. Add data of photo.
41 Should be more reference added, e.g. Farquharson et al., 2018. Temporal and spatial variability in coastline response to declining sea-ice in northwest Alaska. https://doi.org/10.1016/j.margeo.2018.07.007
41-48 Small number of relevant references is used. Should be more reference added.
85-87 This result part looks not logical here after objectives.
102-103 Please exlpain for what reason
134 Data access and filtering (was) were
140 Why to ascending orbit?
Fig.2 Add scale to at least to one figure.
Fig. 2c. "same" - means both?, better ascending and descending
243 0.2 km2, correct upper case
266 MV, explain the abbreviation
318 and Figure 5. Check station coordinates and correct station names on the map.
Should be Prudhoe Bay instead of Nome, Nome instead of Unalakleet, Unlakleet instead of Prudhoe Bay and looks as Kivalina is at Red Dog Dock site.
Scale bar using on figures here and after on similar figure types - it is not clear if 300 km is related to the whole bar or one segment? Please use more clear bar caption.
422-432 Check using correct name stations in the text and on Figure 10.
435-438 This paragraph is not necessary here as this was described in methods
469 Please use correct name of Barents Sea in the text here and after
490 Discussion is better to separate by subsections.
553-576 Should be used more literature for comparison
Author Response
First, we would like to thank the reviewer for taking the time to review our manuscript. We also would like to thank him/her for the positive general evaluation of our manuscript and for considering it to be a well-written and well-structured manuscript with good and clearly presented figures and tables. Thanks to the feedback of the reviewer, we were able to further improve the quality of our manuscript. Please find our detailed answers in the attached pdf.
